# New Bicyclic Azalide Macrolides Obtained by Tandem Palladium Catalyzed Allylic Alkylation/Conjugated Addition Reaction

**DOI:** 10.3390/molecules27020432

**Published:** 2022-01-10

**Authors:** Sulejman Alihodžić, Hana Čipčić Paljetak, Ana Čikoš, Ivaylo Jivkov Elenkov

**Affiliations:** 1Department of Chemistry, Fidelta Ltd. (Selvita Group), Prilaz Baruna Filipovića 29, 10000 Zagreb, Croatia; sulejman.alihodzic@fidelta.eu; 2Center for Translational and Clinical Research, University of Zagreb School of Medicine, Šalata 2, 10000 Zagreb, Croatia; hana.paljetak@mef.hr; 3NMR Center, Ruđer Bošković Institute, Bijenička cesta 54, 10000 Zagreb, Croatia; ana.cikos@irb.hr

**Keywords:** macrolides, azalides, bicyclolides, tandem reaction, allylation, NMR spectroscopy, conformational analysis

## Abstract

Unprecedented tandem allylic alkylation/intermolecular Michael addition was used in the preparation of novel bicyclic azalides. NMR spectroscopy was used not only to unambiguously determine and characterize the structures of these unexpected products of chemical reaction but also to investigate the effect the rigid bicyclic modification has on the conformation of the whole molecule. Thus, some of the macrolides prepared showed antibacterial activity in the range of well-known antibiotic drug azithromycin.

## 1. Introduction

Polyketide macrolides are widely distributed natural products many of which became important medicines in treatment of infectious and neoplastic conditions. In most cases, these compounds contain polyhydroxy macrocyclic lactone glycosylated with one or more sugar moieties [1,2,3]. Erythromycin, a typical representative of this group, was discovered in the 1950s and quickly became commonly used agent in treatment of respiratory and soft tissue infections (Figure 1). Issues with fast growing bacterial resistance, acid instability, and gastro-intestinal side effects led to developments of new, improved, semi-synthetic analogues. Thus, clarithromycin [4] and azithromycin [5] were discovered in the 1980s, followed by ketolides a decade later [6]. Furthermore, linking two or three suitably oriented functionalities by a bridge allowed formation of conformationally rigid condensed polycyclic structures [7] known as bicyclolides [8,9,10,11]. Today, the macrolides are studied, or have already found application, as immunomodulators [12], anti-parasitic agents [13,14], or even in treatment of COVID-19 [15]. Some of them even found application in non-medicinal environment, as chiral selectors in liquid chromatography and capillary electrophoresis [16].

In the course of our search for macrolides with novel biological profiles, we investigated the possibilities for regioselective introduction of (hetero)aryl moiety at position 9a in azalide molecule via C2-4-spacer. Soon, it became obvious that presence and position of the heteroatom in the aromatic system had a profound effect on the outcome of the reaction. Herein, we report unprecedented, highly diastereoselective one-pot-two-step preparation of bicyclic azalides, which are products of tandem palladium, catalyzed allylation/intermolecular conjugate addition reaction.

## 2. Results and Discussion

We envisioned activated 3-quinolyl-allyl alcohols as suitable building blocks for our target molecules, that were to contain quinoline units bound to 9a position of the azalide via C3-linker. Quinolyl analogues of cinnamic alcohol are easily obtainable and excellent substrates for palladium catalyzed allylation of amines and alcohols, a method widely used in total and new chemical entities (NCE) syntheses [17,18]. We started our investigations with the azalide **1**, a precursor in the synthesis of azithromycin and 3-quinolyl-derivative **2**, an intermediate in preparation of cethromycin [19]. The reaction was highly chemo- and stereoselective, affording **3** in 87% yields as a *trans* isomer only (Figure 1).

Nucleophilic addition of the amine to the π-allyl-palladium complex seems to be much slower than its π-σ-π-interconversion, thus allowing complete transfer of *cis*-alkene geometry of 2 into *trans*-one in **3** [20].

Allyl carbonates **8a** and **8b** were prepared as analogues of **2** in a three-step sequence of Wittig olefination, reduction [21,22], and BoC activation (Figure 2). Allylation of **1** with isomeric **8a** and **8b** proceeded smoothly, and products were isolated in acceptable yields (Figure 3).

The NMR analysis, however, showed that the spectra did not match the expected products **9a** and **9b**. In the ^1^H-NMR, instead of olefin signals, two pairs of diastereotopic protons were observed, as well as the signal at 4.27 ppm, corresponding to a new carbinol center. More thorough analysis of 2D NMR spectra revealed the formation of the bicyclic compounds **10a** and **10b**, evidently products of tandem allylation/conjugated addition reaction. The formation of a six-membered ring perfectly explains the observed downfield shifts of ca 5 ppm for C-11 (74.2 ppm in **3** → 78.9 ppm in **10a** and 79.0 ppm in **10b**), as well as upfield shifts of ca 5 ppm for C-10 (62.0 ppm in **3** → 56.6 ppm in **10a** and 56.7 ppm in **10b**).

A possible rationale for such event is the activation of the allyl double bond by the electron-poor 2- and 4-quinolyl rings. Traces of a base (amines present in the mobile phase used in the purification step) triggered addition of nucleophilic 11-OH group to the proximal activated double bond, affording a new morpholine ring. This annulation was successfully performed in one-pot procedure. Addition of catalytic amounts of DBU after the completion of allylation process gave **10a** and **10b**, again in acceptable yields. Retro-allylation, however, was the main side reaction leading back to **1**.

If it is assumed that cyclization process is a conjugated addition, then the presence of suitably oriented electron-withdrawing group (EWG) in the allyl donor is essential for the success of cyclization. Both 2-and 4-quinolinyl moieties polarize and activate the allyl double bond, thus making it a good acceptor in 1,4 and 1,6-addition, respectively. 3-Quinolyl, on the other hand, does not activate the double bond. Therefore, **3**, as expected, did not form cyclization product, even after prolonged heating in the presence of 1 equivalent of DBU.

Since a new chiral center (C-22) is formed in the course of the reaction, NOE and coupling constant analyses were performed in order to determine its stereochemistry. Possible impact of the newly formed morpholine ring on the conformation of the macrocycle and sugar moieties of **10a** and **10b** (deuterated chloroform) was also investigated. In both compounds, the stereochemistry at C-22 was determined to be *R* using strong NOE interactions between protons H-22/H-11, H-22/6-OH, and 6-OH/H-11 (Figure 2b). Such NOE interactions are possible only if H-22 and H-11 are both pseudo-axial in a chair conformation of the newly formed morpholine ring (Figure 2a). This was corroborated by a large coupling constant between H-22 and H-23ax (10.7 Hz for **10a** and 10.8 Hz for **10b**), as well as small coupling constant between H-22 and H-23eq (2.9 Hz for **10a** and 3.7 Hz for **10b**), as seen in Appendix A. Downfield chemical shift of 6-OH (5.7 ppm) indicates a hydrogen bond with the chiral nitrogen at 9a occupying a pseudo-axial position, as well. Similar observations on 6-OH were reported for other conformationally restricted bicyclic azalides, where *N*-9a and *O*-12 were part of the 1,3-oxazine ring [23].

Previous conformational studies on macrolides have established that the macrolactone can adopt two types of conformations (folded-in and folded-out), distinguished by the outward or inward folding of the C-3 to C-5 region [24,25]. Coupling constants ^3^*J*_H-2,H-3_ in both **10a** and **10b** have a value of ca 9 Hz, and their NOESY spectra display strong H-4/H-11 interactions (Appendix A), which establishes that these bicyclic macrolides in chloroform, unlike monocyclic azithromycin [25], adopt an energetically more favorable folded-out conformation. Both sugar moieties, on the other hand, adopt the natural Everett-Tyler chair conformation [24] and keep their usual orientation with respect to the macrocycle.

In both **10a** and **10b**, the quinolyl moiety exhibits only one strong non-trivial NOE contact (H-3 of the heterocycle with H-13 from the macrocycle), suggesting that this group is positioned above the “western” part of the macrocycle.

We still cannot firmly state what impact double bond geometry in **8** has on the stereoselectivity of the cyclization. We assume however, that faster π-σ-π-interconversion of the π-ally-palladium complex leads to formation of plausible *trans* intermediates **9a** or **9b**. Evidently, the annulation is possible in a conformation where the whole side chain is oriented over the “plane” of the macrolide aglycon with double bond and 11-OH in close proximity. 22*R*-isomer is obviously thermodynamically more stable with an aromatic substituent in pseudo-equatorial position.

Expanding the scope of the reaction, we prepared EWG activated allylic carbonates 14a and 14b in a two-step procedure (Figure 4). Wittig condensation of dimeric hydroxyacetaldehyde (**11**) with commercially available phosphonium ylides **5** and **12** afforded allyl alcohols **13a** [26] and **13b** in excellent yields. Their activation with BoC_2_O afforded carbonates **14a** and **14b** in very good yields, as well. It is worth mentioning that, in the case of **13b**, the olefination had low stereoselectivity that reflected in the 2:3 *trans/cis* mixture in **14b**.

One-pot tandem allylation/Michael addition reaction of these new allyl carbonates with azalide **1** afforded **15a** and **15b** in good yields with the same high stereocontrol (Figure 5). It is worth noting that, even though **14b** was used as a mixture of *cis/trans* isomers (3:2), no change in reaction stereoselectivity was observed. Fast π-σ-π-interconversion before *N*-allylation cannot be excluded (vide supra).

Literature analysis demonstrated that similar bridging of *N*-9a and 11-OH group of the azalides via tandem diallylation of **1** was previously reported [27]. Although the obtained bicyclic products had the same carbon skeleton as the one reported here, the stereoselectivity of double allylation was low, giving mixtures of 1:1–1:2 of the possible diastereoisomers.

Antibacterial activities of the prepared macrolides were tested against a panel of relevant Gram-positive (*Streptococcus pneumoniae*, *Streptococcus pyogenes*, and *Staphylococcus aureus*) and Gram-negative (*Haemophilus influenzae* and *Moraxella catarrhalis*) bacterial respiratory tract pathogens that were either sensitive or resistant to macrolide antibiotics. The results, expressed as minimum inhibitory concentrations (MICs), are shown in Table 1, alongside azithromycin as a comparator. Quinolyl derivatives **10a** and **10b** were fully active against *eryS* Gram-positive species, while **15a** and **15b** were not as potent against *S. aureus*.

These two compounds have similar potency and retain good activity against Gram-negative bacteria, but their activity against erythromycin resistant pathogens is weak. It should be noted, however, that EWG present in these molecules (ester in **15a** and nitrile in **15b**) are excellent handles for further transformations and derivatizations that might lead to improved antibacterial activities. To summarize, attached quinolyl moiety influences the antibacterial properties; thus, overall, **10a** is the most potent compound, with improved activity over Azithromycin, especially regarding iMLS *S. pyogenes* (MICs of 0.5 µg/mL versus 16 µg/mL). However, activity against *M. catarrhalis* (4 µg/mL) remains unsatisfactory.

## 3. Experimental Section

### 3.1. General Methods

All reactions utilizing air- and moisture-sensitive reagents were performed in dried glassware under an atmosphere of dry argon or nitrogen. Commercially available reagents and catalysts were purchased from Sigma-Aldrich (Saint Louis, MO, USA), Fluka (Saint Louis, MO, USA), and Merck (Darmstadt, Germany) and were used without additional purifications. Anhydrous solvents were purchased in Sure Seal bottles from Aldrich and used via standard syringe techniques. Compound 1 was purchased from PLIVA as a hydrate and was azeotropically dried by dissolving in toluene and removal of the solvent under reduced pressure. Compound.2 1 was prepared as previously described.

All solvents used for chromatography were technical grade. For thin-layer chromatography (TLC) analysis, Merck pre-coated glass plates (TLC Silica gel 60 F254) were used. TLC spots were visualized either under UV light (at 254 or 366 nm) or by spraying with 5% ethanolic sulfuric acid and subsequent charring. SPE purifications were performed on commercially available cartridges (LC-Silica Packing) and were purchased from Supelco (Saint Louis, MO, USA). Mixtures of DCM and increasing amounts of 3% methanolic ammonia were used as mobile phase.

All solvents used for NMR sample preparation were purchased from EurIsotop (Saint-Aubin, France). NMR spectra were recorded on Bruker Avance III 600, Bruker Avance DRX500, Bruker Avance AV400, and Bruker Avance DPX300 spectrometers (Billerica, MA, USA), equipped with 5 mm diameter broadband inverse and ^1^H/^13^C dual detection probes with z-gradient accessory. The spectra were acquired using standard Bruker pulse sequences on samples dissolved in deuterated chloroform (*CDCl_3_*) with TMS as the internal standard and at 25 °C. NOESY spectra were obtained with the mixing time of 400 ms.

HRMS spectra were acquired as accurate mass centroided data using a Micromass Q-Tof 2 hybrid quadrupole time-of-flight mass spectrometer, equipped with a Z-spray interface, over a mass range of 100–1100 Da, with a scan time of 0.9 s and an interscan delay of 0.1 s. Reserpine was used as the external mass calibrant ([M+H]+ = 609,2812 *m*/*z*). The Q-Tof 2 mass spectrometer was operated in W reflection mode to give a resolution (FWHM) of 16,000–20,000. Ionization was achieved with a spray voltage of 3.2 kV, a cone voltage of 50 V, with cone and desolvation gas flows of 10–20 and 600 L/h, respectively. The source block and desolvation temperatures were maintained at 120 °C and 250 °C, respectively. The elemental composition was calculated using MassLynx v4.1 for the [M+H]+.

Antimicrobial activity was tested against relevant Gram-positive (*S. pneumoniae*, *S. pyogenes*, and *S. aureus*) and Gram-negative (*H. influenzae*, *M. catarrhalis*) bacterial respiratory tract pathogens that were either sensitive or resistant to macrolide antibiotics, due to expression of efflux pumps (M phenotype) or inducible ribosome methylation (iMLSb phenotype).

MICs were determined by the broth microdilution method [28], except that, in the medium used to grow *Streptococcus* strains, lysed blood was replaced by 5% horse serum. The compounds were dissolved in dimethyl sulfoxide (DMSO) at a concentration of 5 mg/mL and azithromycin was used as control. Bacteria for inoculum preparation were grown on appropriate agar plates (Becton Dickinson, Franklin Lakes, NJ, USA): Columbia agar with 5% sheep blood for *Streptococci* and *M. catarrhalis*, chocolate agar for *H. influenzae*, and Mueller-Hinton agar for *Staphylococci*.

### 3.2. Synthetic Procedures

9-Deoxo-9a-(3-(3-Quinolyl)-2-propenyl)-9a-aza-9a-homoerythromycin A (**3**)

A flame-dried Schlenk tube was charged under argon with Pd_2_(dba)_3_·CHCl_3_ (5.18 mg, 5.00 µmol), DPPB (4.26 mg, 10.00 µmol), and toluene (5 mL). The reaction mixture was stirred at ambient temperature for 10 min. Then, 1 (0.367 g, 0.5 mmol) and 2 (0.185 g, 0.65 mmol) were added, and the reaction mixture was stirred at 80 °C for 1.5 h. After cooling to ambient temperature, it was diluted with water (50 mL). pH was lowered to 2, and layers were separated. pH of the aqueous extract was gently increased to pH 7.4 with 1 pH unit step. An extraction with DCM (30 mL) was performed each time. The combined fractions containing the title product were evaporated to dryness. Crude product was purified by SPE. The title product was obtained as pale beige foam (392 mg, 87%).

^1^H-NMR (400 MHz, *CDCl_3_*) δ: 8.95 (s, 1H), 8.10 (s, 1H), 8.04 (d, *J* = 8.4 Hz, 1H), 7.76 (d, *J* = 8.3 Hz, 1H), 7.63 (t, *J* = 7.6 Hz, 1H), 7.45–7.54 (m, 1H), 6.65–6.75 (m, 1H), 6.58 (d, *J* = 16.1 Hz, 1H), 5.04 (d, *J* = 4.5 Hz, 1H), 4.69 (d, *J* = 8.9 Hz, 1H), 4.43 (d, *J* = 7.3 Hz, 1H), 4.18–4.23 (m, 1H), 4.04–4.14 (m, 1H), 3.97 (dd, *J* = 13.2, 3.8 Hz, 1H), 3.83 (brs, 1H), 3.68 (d, *J* = 6.7 Hz, 1H), 3.45–3.54 (m, 1H), 3.41 (dd, *J* = 14.3, 7.6 Hz, 1H), 3.31 (s, 3H), 3.22 (dd, *J* = 9.2, 7.8 Hz, 1H), 3.04 (t, *J* = 8.7 Hz, 1H), 2.78–2.92 (m, 3H), 2.40–2.52 (m, 1H), 2.35 (d, *J* = 15.1 Hz, 1H), 2.28 (s, 6H), 2.19 (d, *J* = 10.9 Hz, 1H), 1.96–2.16 (m, 3H), 1.80–1.94 (m, 1H), 1.72 (d, *J* = 14.7 Hz, 1H), 1.65 (d, *J* = 11.8 Hz, 1H), 1.60 (dd, *J* = 15.0, 5.2 Hz, 1H), 1.44–1.54 (m, 1H), 1.37–1.43 (m, 1H), 1.34 (d, *J* = 6.4 Hz, 3H), 1.32 (s, 3H), 1.27–1.30 (m, 1H), 1.23 (s, 3H), 1.21 (d, *J* = 7.2 Hz, 3H), 1.18 (d, *J* = 6.2 Hz, 3H), 1.08 (s, 3H), 1.06 (d, *J* = 5.7 Hz, 3H), 0.89 (t, *J* = 7.3 Hz, 3H), 0.85 (d, *J* = 6.9 Hz, 3H); ^13^C-NMR (101 MHz, *CDCl_3_*) δ: 178.2 (C), 149.6 (CH), 147.4 (C), 132.1 (CH), 131.3 (CH), 130.0 (C), 129.1 (CH), 129.0 (CH), 128.6 (CH), 128.1 (C), 127.8 (CH), 126.8 (CH), 103.0 (CH), 95.4 (CH), 83.7 (CH), 78.7 (CH), 78.0 (CH), 77.7 (C), 74.7 (C), 74.4 (CH), 74.2 (C), 72.8 (CH), 70.8 (CH), 68.8 (CH), 65.9 (CH), 65.7 (CH), 65.6 (CH_2_), 62.0 (CH), 53.8 (CH_2_), 49.4 (CH_3_), 45.0 (CH), 41.4 (CH_2_), 41.0 (CH), 40.3 (2xCH_3_), 34.9 (CH_2_), 28.8 (CH_2_), 28.3 (CH), 27.0 (CH_3_), 22.1 (CH_3_), 21.5 (CH_3_), 21.4 (CH_2_), 21.3 (CH_3_), 18.3 (CH_3_), 16.3 (CH_3_), 15.1 (CH_3_), 11.2 (CH_3_), 9.4 (CH_3_), 9.3 (CH_3_); HRMS (ES^+^) Calc. for C_49_H_80_N_3_O_12_ ([M+H]^+^): 902.5742; found: 902.5741.

Methyl 3-(2-quinolynyl)-Acrylate (**6a**)

Compounds **4a** (1.1 g, 7 mmol) and **5** (2.34 g, 7 mmol) were suspended in chloroform (50 mL). The reaction mixture was stirred at ambient temperature overnight. The red solution was extracted with 1N HCl (2 × 20 mL). The combined aqueous extracts were washed with EtOAc (20 mL) and neutralized with 10% NaOH. Extraction with EtOAc (3 × 20 mL) afforded organic solution of the title product that was washed with brine (20 mL), dried over Na_2_SO_4_, and the solvent was evaporated. Thus, the red oil formed was passed through a plug of SiO_2_ (50 g) and eluted with Hexanes:EtOAc (10:1). The title product was obtained as yellowish oil that slowly solidified (1.13 g, 75%).

^1^H-NMR (300 MHz, *CDCl_3_*) δ: 8.18 (d, *J* = 8.4 Hz, 1H), 8.12 (d, *J* = 8.3 Hz, 1H), 7.91 (d, *J* = 15.9 Hz, 1H), 7.82 (d, *J* = 8.2 Hz, 1H), 7.74 (ddd, *J* = 8.4, 7.0, 1.4 Hz, 1H), 7.61(d, *J* = 8.5 Hz, 1H), 7.56 (ddd, *J* = 7.1, 6.9, 1.1 Hz, 1H), 7.01 (d, *J* = 15.9 Hz, 1H), 3.84 (s, 3H).

Methyl 3-(4-quinolynyl)-acrylate (**6b**)

Compounds **4b** (785 mg, 5 mmol) and **5** (1.84 g, 5.5 mmol) were suspended in toluene (15 mL). The reaction mixture was refluxed for two hours until complete conversion occurred. After cooling to room temperature, the red solution was extracted with 1N HCl (2 × 20 mL). The combined aqueous extracts were washed with EtOAc (20 mL) and neutralized with 10% NaOH. Extraction with EtOAc (3 × 20 mL) afforded organic solution of the title product that was washed with brine (20 mL), dried over Na_2_SO_4_, and the solvent was evaporated. Thus, the red oil formed was passed through a plug of SiO_2_ (20 g) and eluted with Hexanes:EtOAc (1:1). The title product was obtained as yellowish oil that slowly solidified (930 mg, 87%).

^1^H-NMR (500 MHz, *CDCl_3_*) δ: 9.21 (d, *J* = 4.5 Hz, 1H), 8.42 (d, *J* = 15.8 Hz, 1H), 8.17 (m, 2H), 7.78 (m, 1H), 7.64 (ddd, *J* = 7.1, 6.8, 1.3 Hz, 1H), 7.55 (d, *J* = 4.5 Hz, 1H), 6.66 (d, *J* = 15.8 Hz, 1H), 3.87 (s, 3H).

3-(4-Quinolyl)-2-propene-1-ol (**7b**) [21]

A two-neck-round-bottom flask under argon was charged with compound **6b** (610 mg, 2.86 mmol) and toluene (15 mL). The reaction mixture was cooled to −78 °C, and DIBAL-H (6.22 mL, 6.29 mmol) was added via syringe for a period of 10 min. After 30 min, conversion of the ester was complete. The reaction mixture was warmed to 0 °C and quenched with diethyl ether, water and 15% sodium hydroxide. It was stirred for 15 min at ambient temperature and MgSO_4_ was added. Inorganics were filtered off and washed with ether (2 × 20 mL). Residue obtained after evaporation of the solvent was purified by SPE with mixtures of DCM 3% methanolic ammonia. The title product was obtained as a off-white crystaline solid (290 mg, 51%).

^1^H-NMR (500 MHz, *CDCl_3_*) δ: 8.67 (d, *J* = 4.6 Hz, 1H), 7.97 (d, *J* = 8.5 Hz, 1H), 7.94 (d, *J* = 8.5 Hz, 1H), 7.55 (t, *J* = 7.2 Hz, 1H), 7.38 (t, *J* = 7.3 Hz, 1H), 7.26 (d, *J* = 4.6 Hz, 1H), 7.22 (d, *J* = 15.6 Hz, 1H), 6.45 (dt, *J* = 15.6, 4.9 Hz, 1H), 4.36 (dd, *J* = 4.9, 1.5 Hz, 2H), 3.11 (brs., 1H); ^13^C-NMR (126 MHz, *CDCl_3_*) δ: 149.9, 148.3, 143.0, 136.2, 129.7, 129.4, 126.5, 126.2, 124.4, 123.6, 117.5, 63.0.

3-(2-Quinolinyl)-2-propene-1-ol (**7a**) [22].

Compound was prepared analogously to **7b**, starting with **6a** (770 mg, 3.61 mmol). Crude product was purified by SPE on and eluted with mixtures of hexane and ethyl acetate. The product was obtained as a pale green solid (160 mg, 27%).

^1^H-NMR (300 MHz, *CDCl_3_*) δ: 8.07 (d, *J* = 8.4 Hz, 1H), 8.06 (d, *J* = 8.5 Hz, 1H), 7.75 (d, *J* = 8.2 Hz, 1H), 7.65-7.72 (m, 1H), 7.43-7.56 (m, 2H), 7.04 (d, *J* = 16.0 Hz, 1H), 6.91 (dt, *J* = 16.0, 4.5 Hz, 1H), 4.48 (dd, *J* = 4.4, 1.1 Hz, 2H); ^13^C-NMR (75 MHz, *CDCl_3_*) δ: 155.1, 147.0, 135.9, 135.8, 129.5, 129.1, 128.2, 126.8, 126.6, 125.5, 118.3, 62.0.

3-(4-Quinolyl)-allyl tert-butyl carbonate (**8b**).

Compound **7b** (280 mg, 1.51 mmol) was suspended in DCM (5 mL). Boc_2_O (0.42 mL, 1.81 mmol) and DMAP (18.47 mg, 0.151 mmol) were added, and the reaction mixture Was stirred at ambient temperature for 1 h. The reaction was quenched with sat NaHCO_3_ (10 mL) and stirred for 10 min more. Layers were separated, and the aqueous one was extracted with diethyl ether (3 × 10 mL). Organics were dried over Na_2_SO_4_ and evaporated to dryness. Residue was purified by SPE and eluted with Hex:EtOAc. The title compound was obtained as colorless oil (340 mg, 71%).

^1^H-NMR (500 MHz, *CDCl_3_*) δ: 8.88 (d, *J* = 4.5 Hz, 1H), 8.08-8.12 (m. 2H), 7.70-7.75 (m, 1H), 7.55-7.59 (m, 1H), 7.44 (d, *J* = 4.5 Hz, 1H), 7.38 (dt, *J* = 15.8, 1.3 Hz, 1H), 6.50 (dt, *J* = 15.8, 5.5 Hz, 1H), 4.91(dd, *J* = 5.5, 1.3 Hz, 1H), 1.54 (s, 9H); ^13^C-NMR (101 MHz, *CDCl_3_*) δ: 153.2, 150.2, 148.5, 142.1, 130.1, 130.0, 129.4, 128.3, 126.6, 126.1, 123.5, 117.8, 82.6, 66.8, 27.8 (3C).

3-(2-Quinolyl)-allyl tert-butyl carbonate (**8a**).

Compound **7a** (150 mg, 0.810 mmol) was solved in DCM (5 mL). Boc_2_O (0.226 mL, 0.972 mmol) and DMAP (9.89 mg, 0.081 mmol) were added, and the reaction mixture was stirred at ambient temperature for 0.5 h. The reaction was quenched with sat NaHCO_3_ (10 mL) and stirred for 10 min more. Layers were separated, and the aqueous one was extracted with diethyl ether (3 × 10 mL). Organics were dried over Na_2_SO_4_ and evaporated to dryness. Residue was purified by SPE and eluted with Hex:EtOAc. The title compound was obtained as colorless oil (153 mg, 66%).

^1^H-NMR (400 MHz, *CDCl_3_*) δ: 8.08 (d, *J* = 8.5 Hz, 1H), 8.02 (dd, *J* = 8.3, 0.8 Hz, 1H), 7.75 (dd, *J* = 8.1, 1.1 Hz, 1H), 7.67 (m, 1H), 7.51 (d, *J* = 8.7 Hz, 1H), 7.47 (dd, *J* = 6.9, 1.2 Hz, 1H), 6.95 (dt, *J* = 16.1, 1.3 Hz, 1H), 6.85 (dt, *J* = 16.0, 5.5 Hz, 1H), 4.83 (dd, *J* = 5.6, 1.3 Hz, 2H), 1.50 (s, 9H); ^13^C-NMR (101 MHz, *CDCl_3_*) δ: 154.9, 153.3, 148.0, 136.4, 133.7, 129.7, 129.3, 127.4, 127.0, 126.4, 119.1, 82.4, 66.7, 27.8 (3C).

Methyl 4-hydroxycrotonate (**13a**) [26]

Compounds **5** (6.68 g, 20 mmol) and **11** (1.2 g, 10 mmol) were suspended in chloroform (60 mL). The reaction mixture was stirred at ambient temperature for 24 h. The solvent was removed under reduced pressure, and residue was subjected to bulb-to-bulb distillation (150 °C/1.2 × 10^−1^ mbarr). The product was obtained as colorless oil (2.1 g, 88%).

^1^H-NMR (400 MHz, *CDCl_3_*) δ: 7.05 (dt, *J* = 4.0, 15.6 Hz, 1H), 6.15-6.08 (m, 1H), 4.38-4.33 (m, 2H), 3.74 (s, 3H).

4-Hydroxycrotononitrile (**13b**)

Compound was prepared the same way as **13a**, starting with compounds **12** (7.88 g, 26.2 mmol) and **11** (1.57 g, 13.08 mmol). The product was obtained as colorless oil (4:1 *trans/cis* mixture) after a bulb-to-bulb distillation (120 °C/1.5 × 10^−1^ mbarr). Yield (1.94 g, 89%).

^1.^ H NMR (500 MHz, *CDCl_3_*) δ: (*trans* isomer) 6.84 (dt, J = 16.3, 3.4 Hz, 1H), 5.73 (dt, J = 16.4, 2.3 Hz, 1H), 4.33 (t, J = 2.9 Hz, 2H); (*cis* isomer) 6.62 (dt, J = 11.4, 5.8 Hz, 1H), 5.44 (dt, J = 11.3, 1.7 Hz, 1H), 4.47 (dd, J = 5.8, 1.5 Hz, 2H); ^13^C-NMR (126 MHz, *CDCl_3_*) δ: (*trans* isomer) 153.6, 117.3, 98.4, 61.5; (*cis* isomer) 152.9, 117.3, 99.4, 61.2.

Methyl-4-tert-butyloxycarbonyloxy-crotonate (**14a**)

Compound **13a** (6.1 g, 52.5 mmol) was solved in DCM (70 mL). Boc_2_O (14.64 mL, 63.0 mmol) was added, followed by DMAP (0.321 g, 2.63 mmol). The reaction mixture was stirred for 2 h at room temperature and then washed with 1N HCl (2 × 50 mL), water (50 mL), and brine (50 mL). Organic layer was dried over Na_2_SO_4_, and the solvent removed under reduced pressure. The red-brown oily residue was passed through a plug of SiO_2_ (30 g) and eluted with EtOAc:Hex (1:2). The title product was obtained as yellow oil (10 g, 88%).

^1^H-NMR (600 MHz, *CDCl_3_*) δ: 6.90 (td, *J* = 15.7, 4.7 Hz, 1H), 6.02 (dt, *J* = 15.9, 1.9 Hz, 1H), 4.68 (dd, *J* = 4.6, 2.0 Hz, 2H), 3.70 (s, 3H), 1.45 (s, 9H); ^13^C-NMR (151 MHz, *CDCl_3_*) δ: 166.2, 152.9, 141.3, 121.8, 82.7, 64.8, 51.7, 27.7.

4-*tert*-Butyloxycarbonyloxy-crotononitrile (**14b**)

Compound was prepared in a similar way to **14a** starting from compound **13b** (1.32 g, 15.9 mmol). The title product was obtained as a yellowish oily mixture of *trans/cis* isomers(4:1) (2.4 g, 78%).

^1^H-NMR (300 MHz, *CDCl_3_*) δ: (*trans* isomer) 6.68 (dt, *J* = 16.4, 4.4 Hz, 1H), 5.59 (dt, *J* = 16.3, 2.2 Hz, 1H), 4.65 (dd, *J* = 4.4, 2.1 Hz, 2H), 1.44 (s, 9H); (*cis* isomer) 6.54 (dt, *J* = 11.2, 6.2 Hz, 1H), 5.50 (dt, *J* = 11.2, 1.7 Hz, 1H), 4.84 (dd, *J* = 6.1, 1.7 Hz, 2H), 1.47 (s, 9H); ^13^C-NMR (75 MHz, *CDCl_3_*) δ: (*trans* isomer) 152.5, 147.5, 116.5, 101.1, 83.3, 64.5, 27.6 (3C).

(22R)-9-Deoxo-11-deoxy-11-9a-(epoxyethano)-22-(2-quinolyl)methyl-9a-aza-9a-homoerythromycin A (**10a**)

A flame dried Schlenk tube was charged under argon with compounds **1** (184 mg, 0.25 mmol), **8a** (85 mg, 0.27 mmol) and dry toluene (2.5 mL). Pd_2_(dba)_3_·CHCl_3_ (6 mg, 0.005 mmol) and DPPB (5 mg, 0.01 mmol) were added, and the reaction mixture was heated at 80 °C for 1 h. After completion of the reaction, DBU (152 mg, 1 mmol) was added, and the reaction mixture was heated for another 1 h. The solvent was evaporated under reduced pressure, and foamy residue was chromatographed on SiO_2_ (20 g) EtOAc:Hexanes:DEA (100:100:10) and further on the same amount of silica with DCM:MeOH:NH_4_OH (90:6:0.5). The title product was obtained as a white solid (80 mg, 35%).

^1^H-NMR (500 MHz, *CDCl_3_*) δ: 8.12 (d, *J* = 8.5 Hz, 1H), 8.04 (d, *J* = 8.5 Hz, 1H), 7.78 (dd, *J* = 8.4, 1.1 Hz, 1H), 7.75 (d, *J* = 8.5 Hz, 1H), 7.63-7.71 (m, 1H), 7.49 (ddd, *J* = 8.0, 6.9, 1.2 Hz, 1H), 5.94 (s, 1H), 4.97 (dd, *J* = 4.6, 1.5 Hz, 1H), 4.67 (dd, *J* = 11.3, 2.1 Hz, 1H), 4.48 (d, *J* = 7.3 Hz, 1H), 4.28-4.36 (m, 1H), 4.25 (dd, *J* = 8.5, 1.8 Hz, 1H), 4.09 (dq, *J* = 8.7, 6.3 Hz, 1H), 3.77 (d, *J* = 6.7 Hz, 1H), 3.51-3.60 (m, 1H), 3.35 (d, *J* = 1.8 Hz, 1H), 3.32 (s, 3H), 3.30-3.34 (m, 1H), 3.27 (dd, *J* = 15.0, 4.9 Hz, 1H), 3.22 (dd, *J* = 15.0, 8.2 Hz, 1H), 3.07 (d, *J* = 8.5 Hz, 1H), 2.87-2.94 (m, 1H), 2.59-2.71 (m, 3H), 2.54 (t, *J* = 11.0 Hz, 1H), 2.44 (s, 6H), 2.34 (dd, *J* = 15.0, 1.8 Hz, 1H), 2.29 (dd, *J* = 12.7, 3.2 Hz, 1H), 1.93-2.04 (m, 2H), 1.82-1.90 (m, 3H), 1.72 (dd, *J* = 14.6, 1.8 Hz, 1H), 1.65-1.70 (m, 1H), 1.63 (dd, *J* = 15.0, 4.9 Hz, 1H), 1.36-1.45 (m, 2H), 1.35 (d, *J* = 6.4 Hz, 3H), 1.33 (s, 3H), 1.28-1.32 (m, 1H), 1.26 (s, 3H), 1.25 (d, *J* = 7.3 Hz, 3H), 1.24 (d, *J* = 6.1 Hz, 3H), 1.07 (d, *J* = 7.6 Hz, 3H), 1.01 (s, 3H), 0.93 (d, *J* = 6.4 Hz, 3H), 0.91 (d, *J* = 7.0 Hz, 3H), 0.70 (t, *J* = 7.5 Hz, 3H); ^13^C-NMR (75 MHz, *CDCl_3_*) δ: 175.9 (C), 158.7 (C), 147.6 (C), 136.1 (CH), 129.2 (CH), 128.7 (CH), 127.5 (CH), 127.0 (C), 125.8 (CH), 122.1 (CH), 103.1 (CH), 96.5 (CH), 84.9 (CH), 80.6 (CH), 78.9 (CH), 77.8 (CH), 77.4 (CH), 74.1 (CH), 73.9 (C), 73.1 (C), 72.9 (C), 71.1 (CH), 68.6 (CH), 66.2 (CH), 65.6 (CH), 64.1 (CH_2_), 56.6 (CH), 49.4 (CH_2_), 49.4 (CH_3_), 44.8 (CH), 42.7 (CH_2_), 41.5 (CH_2_), 40.4 (2xCH_3_), 39.6 (CH_3_), 35.3 (CH_2_), 29.6 (CH_2_), 26.6 (C & CH_3_), 21.6 (CH_3_), 21.3 (CH_3_), 21.3 (CH_3_), 20.2 (CH_2_), 18.6 (CH_3_), 16.5 (CH_3_), 15.3 (CH_3_), 10.7 (CH_3_), 9.6 (CH_3_), 5.4 (CH_3_); HRMS (ES^+^) Calc. for C_49_H_80_N_3_O_12_ ([M+H]^+^): 902.5742; found: 902.5740.

(22R)-9-Deoxo-11-deoxy-11-9a-(epoxyethano)-22-(4-quinolyl)methyl-9a-aza-9a-homoerythromycin A (**10b**)

A flame dried Schlenk tube was charged under argon with compounds **1** (184 mg, 0.25 mmol), **8b** (77 mg, 0.27 mmol), and dry toluene (3 mL). Pd_2_(dba)_3_·CHCl_3_ (6 mg, 0.005 mmol) and DPPB (5 mg, 0.01 mmol) were added, and the reaction mixture was heated at 80 °C for 1 h. After completion of the reaction, DBU (152 mg, 1 mmol) was added, and the reaction mixture was heated for another 1 h. The solvent was evaporated under reduced pressure, and foamy residue was chromatographed on SiO_2_ (20 g) EtOAc:Hexanes:DEA (100:100:10) and further on the same amount of silica with DCM:MeOH:NH_4_OH (90:6:0.5). The title product was obtained as a white solid (103 mg, 46%).

^1^H-NMR (500 MHz, *CDCl_3_*) δ: 8.84 (d, *J* = 4.3 Hz, 1H), 8.11 (d, *J* = 7.9 Hz, 1H), 8.06 (d, *J* = 8.1 Hz, 1H), 7.68 (m, 2H), 7.55 (ddd, *J* = 7.9, 7.0, 1.1 Hz, 1H), 6.58 (s, 1H), 4.94 (dd, *J* = 4.4, 2.5 Hz, 1H), 4.60 (dd, *J* = 11.1, 2.2 Hz, 1H), 4.51 (d, *J* = 7.2 Hz, 1H), 4.27 (m, 1H), 4.17 (dd, *J* = 9.0, 1.6 Hz, 1H), 4.06 (dq, *J* = 8.2, 6.4 Hz, 1H), 3.78 (d, *J* = 6.6 Hz, 1H), 3.58 (m, 1H), 3.37 (dd, *J* = 9.8, 7.6 Hz, 1H), 3.34 (d, *J* = 1.8 Hz, 1H), 3.31 (s, 3H), 3.30 (d, *J* = 13.7 Hz, 1H), 3.23 (dd, *J* = 14.5, Hz, 1H), 3.09 (d, *J* = 8.1 Hz, 1H), 2.90 (dq, *J* = 8.9, 7.2 Hz, 1H), 2.87 (m, 1H), 2.65 (s, 6H), 2.64 (m, 2H), 2.52 (d, *J* = 11.3 Hz, 1H), 2.41 (m, 1H), 2.30 (dd, *J* = 14.7, 2.3 Hz, 1H), 2.29 (dd, *J* = 12.7, 3.3 Hz, 1H), 2.24 (s, 1H), 2.02 (m, 3H), 1.89 (m, 1H), 1.68 (dd, *J* = 14.7, 1.2 Hz, 1H), 1.64 (dd, *J* = 14.9, 4.6 Hz, 1H), 1.62 (m, 1H), 1.37 (m, 3H), 1.34 (d, *J* = 7.0 Hz, 3H), 1.31 (s, 3H), 1.25 (d, *J* = 6.8 Hz, 6H), 1.27 (s, 3H), 1.04 (d, *J* = 7.5 Hz, 3H), 0.98 (s, 3H), 0.92 (d, *J* = 7.0 Hz, 3H), 0.75 (t, *J* = 7.4 Hz, 3H);^13^C-NMR (75.47 MHz, *CDCl_3_*) δ: 174.5 (C), 149.2 (CH), 147.0 (C), 142.5 (C), 129.2 (CH), 127.8 (CH), 127.5 (C), 125.1 (CH), 122.7 (CH), 120.8 (CH), 101.9 (CH), 95.7 (CH), 84.7 (CH), 80.1 (CH), 77.9 (CH), 76.4 (CH), 76.2 (CH), 72.9 (CH), 72.7 (C), 72.0 (C), 72.0 (C), 70.0 (CH), 67.2 (CH), 65.7 (CH), 64.7 (CH), 63.0 (CH_2_), 55.5 (CH), 48.4 (CH_2_), 48.3 (CH_3_), 43.5 (CH), 40.3 (CH_2_), 39.6 (2xCH_3_), 37.8 (CH), 34.4 (CH_2_), 33.7 (CH_2_), 30.3, (CH_2_), 25.6 (CH), 25.4 (CH_3_), 20.5 (CH_3_), 20.3 (CH_3_), 20.1 (CH_3_), 19.1 (CH_2_), 17.5 (CH_3_), 15.6 (CH_3_), 14.1 (CH_3_), 9.7 (CH_3_), 8.7 (CH_3_), 4.1 (CH_3_); HRMS (ES^+^) Calc. for C_49_H_80_N_3_O_12_ ([M+H]^+^): 902.5742; found: 902.5753.

(22R)-9-Deoxo-11-deoxy-11-9a-(epoxyethano)-22-(methoxycarbonyl)methyl-9a-aza-9a-homoerythromycin A (**15a**)

According to the procedure above, starting with compounds **1** (367 mg, 0.5 mmol) and **14a** (130 mg, 0.6 mmol). Yield (170 mg, 41%).

^1^H-NMR (500 MHz, *CDCl_3_*) δ: 5.61 (s, 1H), 4.95 (d, *J* = 4.3 Hz, 1H), 4.88 dd, *J* = 11.3, 1.8 Hz, 1H), 4.43 (d, *J* = 7.0 Hz, 1H), 4.23 (dd, *J* = 8.8, 1.8 Hz, 1H), 3.98–4.14 (m, 2H), 3.75 (d, *J* = 6.7 Hz, 1H), 3.70 (s, 3H), 3.45–3.57 (m, 1H), 3.35 (d, *J* = 1.8 Hz, 1H), 3.31 (s, 3H), 3.23 (dd, *J* = 10.1, 7.3 Hz, 1H), 3.03 (t, *J* = 8.5 Hz, 1H), 2.88–2.97 (m, 1H), 2.67 (qd, *J* = 6.4, 1.2 Hz, 1H), 2.60 (dd, *J* = 15.0, 2.8 Hz, 1H), 2.59 (dd, *J* = 14.6, 5.2 Hz, 1H), 2.53 (d, *J* = 11.3 Hz, 1H), 2.48 (dd, *J* = 14.8, 6.9 Hz, 1H), 2.41–2.46 (m, 1H), 2.36 (d, *J* = 15.0 Hz, 1H), 2.29 (s, 6H), 2.24–2.27 (m, 2H), 1.97–2.03 (m, 1H), 1.92–1.97 (m, 1H), 1.80–1.92 (m, 2H), 1.76 (dd, *J* = 14.6, 1.5 Hz, 1H), 1.63–1.69 (m, 1H), 1.60 (dd, *J* = 15.0, 4.9 Hz, 1H), 1.44–1.49 (m, 1H), 1.41 (dd, *J* = 14.8, 4.7 Hz, 1H), 1.33 (s, 3H), 1.32 (d, *J* = 6.6 Hz, 3H), 1.26–1.31 (m, 1H), 1.24 (s, 3H), 1.23 (d, *J* = 6.0 Hz, 3H), 1.23 (d, *J* = 6.1 Hz, 3H), 1.09 (d, *J* = 7.6 Hz, 3H), 1.04 (s, 3H), 1.00 (d, *J* = 6.4 Hz, 3H), 0.92 (d, *J* = 7.0 Hz, 3H), 0.85 (t, *J* = 7.9 Hz, 3H); ^13^C-NMR (126 MHz, *CDCl_3_*) δ: 175.8 (C), 170.9 (C), 103.4 (CH), 96.7 (CH), 84.7 (CH), 80.7 (CH), 79.0 (CH), 78.2 (CH), 77.1 (CH), 74.0 (C), 73.2 (C), 72.9 (C), 72.4 (CH), 71.2 (CH), 68.8 (CH), 65.9 (CH), 65.7 (CH), 64.1 (CH), 56.5 (CH), 51.7 (CH_3_), 49.5 (CH_3_), 48.9 (CH_2_), 44.8 (CH), 41.7 (CH_2_), 40.5 (2xCH_3_), 39.7 (CH), 38.7 (CH_2_), 35.4 (CH_2_), 29.0 (CH_2_), 26.8 (CH), 26.7 (CH_3_), 21.6 (CH_3_), 21.5 (CH_3_), 21.4 (CH_3_), 20.4 (CH_2_), 18.7 (CH_3_), 16.7 (CH_3_), 15.3 (CH_3_), 10.8 (CH_3_), 9.7 (CH_3_), 5.5 (CH_3_); HRMS (ES^+^) Calc. for C_42_H_77_N_2_O_14_ ([M+H]^+^): 833.53; found: 833.5382.

(22R)-9-Deoxo-11-deoxy-11-9a-(epoxyethano)-22-(cyano)methyl-9a-aza-9a-homoerythromycin A (**15a**)

According to the procedure above, starting with compounds **1** (735 mg, 1 mmol, **14b** (217 mg, 1.18 mmol), yield was 621 mg (69%).

^1^H-NMR (500 MHz, *CDCl_3_*) δ: 5.54 (s, 1H), 4.90–4.98 (m, 2H), 4.41 (d, *J* = 7.3 Hz, 1H), 4.19 (dd, *J* = 8.7, 1.4 Hz, 1H), 4.06 (dq, *J* = 9.2, 6.2 Hz, 1H), 3.90–4.00 (m, 1H), 3.74 (d, *J* = 7.6 Hz, 1H), 3.46–3.56 (m, 1H), 3.40 (d, *J* = 1.53 Hz, 1H), 3.31 (s, 3H), 3.23 (dd, *J* = 10.1, 7.3 Hz, 1H), 3.03 (d, *J* = 9.2 Hz, 1H), 2.89–2.99 (m, 1H), 2.79 (dd, *J* = 17.1, 4.9 Hz, 1H), 2.70–2.74 (m, 1H), 2.65–2.70 (m, 1H), 2.60 (dd, *J* = 11.6, 3.1 Hz, 1H), 2.51 (dd, *J* = 17.1, 4.0 Hz, 1H), 2.41–2.48 (m, 1H), 2.36 (d, *J* = 15.5 Hz, 1H), 2.31–2.33 (m, 1H), 2.29 (s, 6H), 2.03 (t, *J* = 12.2 Hz, 1H), 1.82–1.96 (m, 3H), 1.77 (d, *J* = 15.0 Hz, 1H), 1.63–1.70 (m, 1H), 1.60 (dd, *J* = 15.3, 4.9 Hz, 1H), 1.46–1.55 (m, 1H), 1.42 (dd, *J* = 15.0, 4.9 Hz, 1H), 1.34 (s, 3H), 1.32 (d, *J* = 6.1 Hz, 3H), 1.26–1.31 (m, 1H), 1.24 (s, 3H), 1.23 (d, *J* = 6.4 Hz, 3H), 1.23 (d, *J* = 6.1 Hz, 3H), 1.10 (d, *J* = 7.3 Hz, 3H), 1.07 (d, *J* = 7.0 Hz, 3H), 1.06 (s, 3H), 0.93 (d, *J* = 7.0 Hz, 3H), 0.88 (t, *J* = 7.3 Hz, 3H); ^13^C-NMR (126 MHz, *CDCl_3_*) δ: 176.1 (C), 116.5 (C), 103.4 (CH), 96.8 (CH), 84.4 (CH), 80.7 (CH), 79.3 (CH), 78.1 (CH), 77.1 (CH), 74.1 (C), 73.2 (C), 72.8 (C), 71.1 (CH), 70.1 (CH), 68.8 (CH), 65.8 (CH), 65.6 (CH), 64.0 (CH_2_), 56.4 (CH), 49.4 (CH_3_), 48.1 (CH_2_), 44.8 (CH), 41.7 (CH_2_), 40.4 (2 × CH_3_), 39.7 (CH), 35.4 (CH_2_), 28.9 (CH_2_), 26.7 (CH), 26.7 (CH_3_), 22.3 (CH_2_), 21.6 (CH_3_), 21.4 (CH_3_), 21.3 (CH_3_), 20.3 (CH_2_), 18.7 (CH_3_), 16.7 (CH_3_), 15.3 (CH_3_), 10.7 (CH_3_), 9.6 (CH_3_), 5.7 (CH_3_); HRMS (ES^+^) Calc. for C_41_H_74_N_3_O_12_ ([M+H]^+^): 800.5273; found: 800.5266.

## 4. Conclusions

In conclusion, a tandem one-pot palladium catalyzed allylic alkylation/intermolecular Michael addition procedure was developed. It allowed preparation of a series of bicyclic azalides that contained a condensed, highly substituted morpholine ring created by bridging of azalide *N*-9a and 11-OH group via tandem diallylation. This modification, although positioned in the western part of the macrocycle, affected the conformation of the whole molecule, causing the outward folding of the C-3 to C-5 region into a so-called “folded-out” conformation. Although a similar reaction with similar resulting compound was reported earlier [19], stereoselectivity of our procedure is unprecedented, yielding only one stereoisomer in all obtained products. Although compounds containing heteroaryl substituents seem to be potent growth inhibitors of erythromycin sensitive strains, they still have low activity against resistant strains. Therefore, we suggest employing them as the starting points for further transformations and optimizations.

## Data Availability

The data presented in this study are available in experimental section of the article or in Appendix A.

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
