# Peer review of "New Bicyclic Azalide Macrolides Obtained by Tandem Palladium Catalyzed Allylic Alkylation/Conjugated Addition Reaction"

_molecules, 2022, doi:10.3390/molecules27020432_

Round 1

Reviewer 1 Report

In this manuscript, the authors conveyed an overview of one-pot-two-step preparation of new bicyclic azalides via unprecedented highly diastereoselective tandem allylic alkylation/intermolecular conjugate addition. As well as, the express that some of these macrolides thus made exhibit the antibacterial activity in the range of well-known antibiotic drug azithromycin. Overall, in my view these modified azalide macrolides might be biologically worthy and I recommend its acceptance for publication in this Journal after authors amend some concerns which are appended below.

Comments

  1. The numbering of formulas in the Schemes are confused. Why compound 4a and 4b appears just after the compound 8a/8b
  2. It is recommend in line 12/page 1, better to use the word new instead of novel
  3. Some editorial errors, for example in Scheme 3, space in temp must be checked/page 5 LINE 94 remove italics CDCl3
  4. In Scheme 3, page 4, is there any change, if you switch base instead of DBU???
  5. In experimental compound must be bold font
  6. In SI file, compound 15a /10a/4a/4b is it 13C or DEPT/APT??? and no peak values for 15a, b
  7. From SI file, for example, Compound 4b NMR data not consistent with the provided spectra, it is suggested please check these errors throughout the manuscript.

Author Response

We are very thankful to the remarks made by the reviewer. Our answers are bellow in red:

Comments

  1. The numbering of formulas in the Schemes are confused. Why compound 4a and 4b appears just after the compound 8a/8b
    We accepted the remark and changed the numbering in order to prevent misleading of the readers. Thus compounds 6 turned to 4; 7--> 6; 8--> and 4--> 8. All those changed were applied on schemes 2 and 3 and in the text on pages3 (row 60 & 62); 44 (row 66); 6 (row 70); . These numbering changes also appear in IS in Table S-1, Figures S3-S6.
  2. It is recommend in line 12/page 1, better to use the word new instead of novel
    We thank to the reviewer for the suggestion but we prefer to stay on the original text because „novel“ is quite often used in the literature when describing preparation of new group of compounds e.g. see the title of REF. 6a, 6c, 6d and 15.
  3. Some editorial errors, for example in Scheme 3, space in temp must be checked/page 5 LINE 94 remove italics CDCl3
    We accepted and introduced the space between value and temperature sign on Scheme 3. We made analogous corrections on Schemes 2 and 4. We changed „CDCl3“ in the Results and discussion section with „deuterated chloroform“. In the general part of the Experimental section we added „deuterated chloroform (CDCl3)“ (page 10, row 178).
  4. In Scheme 3, page 4, is there any change, if you switch base instead of DBU???
    We agree that it is possible to achieve similar transformation by using other bases (we suspect that DEA caused cyclisation first time in the isolation of 10a) but in our case we tried one-pot protocol with DBU only and that is the reason it is specifically stated on schemes 3 and 4. We believe that we have to keep it as in the original text.
  5. In experimental compound must be bold font
    Reviewer's remark was taken into consideration and all compound numbers in the experimental section were put in bold.
  6. In SI file, compound 15a /10a/4a/4b is it 13C or DEPT/APT??? and no peak values for 15a, b
    All the 13C-spectra included in Si are DEPT. The only exclusions ore spectra of compounds 15a and 15b that are APT. That additional information was introduced in the test in Table S-1 and bellow each 13C spectrum on pages 3, 5, 7, 9, 12, 17 and 19.
  7. From SI file, for example, Compound 4b NMR data not consistent with the provided spectra, it is suggested please check these errors throughout the manuscript.
    We thank to the reviewer for the thorough analysis. In the experimental section, we corrected the erroneous data (compound is now 8b,vide supra) according to the presented spectrum (page 16, row 303).

Reviewer 2 Report

The paper is of interest in the field of Medicinal Chemistry. However there are some comments before the paper can be accepted:

Comments: 1- Polyketide macrolides are widely distributed natural products many of which be-21 came important medicines in treatment of infectious and neoplastic conditions...........required recent ref

2- The sequence of preparation couldnt easy understand. After Scheme 1. The authors started with compounds numbering as 6a and 6b to prepare compounds 4a and 4b. The sequence of numbering let the authors not simply follow up the reactions. As after Scheme 1, only numbering should be started by 4a and 4b for compounds 6a and 6b. Also, the structure of compounds 4a and 4b  should be replaced by compounds 6a and 6b. Accordingly and in Scheme 3, reaction of 1 should be with compounds 6a and 6b.

3- nOe should be NOE

4- Why the authors didnt separte cis/trans mixture of compounds 14a and 14b and its better to do that for compounds 13a and 13b. It would then useful to study the steroselectivity during preparation of compounds 15a and 15b.

5- The numbers of compounds in the experimental section should be in bold fonts.

6- mass error -0.2 ppm......there is no need. The same should be omitted for all compounds.

7- SiO2 was wrongly wrritten.

8- tert or t should be wrritten in one style.

9- Solvents should be wrritten in one style...for example chloroform (written as CHCl3 and chloroform).

10- Cis- and trans should be in italic forms.

11- I didnt find the antibiotic reference. It might be AZM....what is this.

12- A large distance was found in UV light ( 254 or 366 nm)

13- A concise experimental for the biology has to be written

Author Response

We thank to the reviewer for the very instructive comments and our answers are stated bellow in red.

Comments: 1- Polyketide macrolides are widely distributed natural products many of which be-21 came important medicines in treatment of infectious and neoplastic conditions...........required recent ref
We believe that review articles cited in Ref 1 are representative sources for readers. Some of them quite recent (e.g .ref. 1c). If the edditor however, suggests to introduce new references we will do so.

2- The sequence of preparation couldnt easy understand. After Scheme 1. The authors started with compounds numbering as 6a and 6b to prepare compounds 4a and 4b. The sequence of numbering let the authors not simply follow up the reactions. As after Scheme 1, only numbering should be started by 4a and 4b for compounds 6a and 6b. Also, the structure of compounds 4a and 4b  should be replaced by compounds 6a and 6b. Accordingly and in Scheme 3, reaction of should be with compounds 6a and 6b.
As this reviewer’s remark is similar to the one of the first reviewer. We made the changes as described previously.
3- nOe should be NOE
We accepted reviewer's suggestion and changed NOE as all caps in figure 2a and in the text on pages 7 (row 94, 99,100 & 112); 8 (row 123)

4- Why the authors didnt separte cis/trans mixture of compounds 14a and 14b and its better to do that for compounds 13a and 13b. It would then useful to study the steroselectivity during preparation of compounds 15a and 15b.
As it was stated in the text “Nucleophilic addition of the amine to the π-allyl-palladium complex seems to be much slower than its π-σ-π-interconversion, thus allowing complete transfer of cis-alkene geometry of 2 into trans-one in 3”. Because of that we believe that either allyl carbonate isomer after oxidative addition to palladium catalyst will isomerise before reacting with compound 1 and thus will afford “trans” intermediate (e.g. 9a or 9b). Bicyclic products obtained in the next step should be identical. This is the reason why we did not tried to separate mixtures 14a and 14b.

5- The numbers of compounds in the experimental section should be in bold fonts.
We accepted that remark and corrected in the text.

6- mass error -0.2 ppm......there is no need. The same should be omitted for all compounds.
We quoted the mass error for accuracy. However, we accepted the reviewer's suggestion and removed those data in the text of the experimental section (page 10, row 204; page 11, row 245; page 14, rows 384 and 414; page 15, rows 434 and 455).

7- SiO2 was wrongly wrritten.
Correction was performed.

8- tert or t should be wrritten in one style.
“tert” was corrected on pages 12 (row 295 & 309); 13 (rows 338 & 348)

9- Solvents should be written in one style...for example chloroform (written as CHCl3 and chloroform).
We agree with the reviewer. When used as a solvent “chloroform” is stated as a word (page 7. Row 97) and in Scheme 4. When used as part of the precatalyst complex (Pd2(dba)3.CHCl3) it is stated as a formula.

10- Cis- and trans should be in italic forms.
We are thankful for that remark. Changes were done on page 13

11- I didnt find the antibiotic reference. It might be AZM....what is this.
The antibiotic reference is Azithromycin and AZM was changed to Azithromycin in Table 1 (page 9)

12- A large distance was found in UV light ( 254 or 366 nm)
The stated wavelengths (254 and 366) are standard wavelengths on commercial UV lamps most often used for visualisation of UV-active chromatographic spots.

13- A concise experimental for the biology has to be written.
We apologise for the lack of data for biological experiments. They were omitted by accident in the course of manuscript preparation. Now these data are included at pages 10 and 11.

Round 2

Reviewer 2 Report

The paper could be accepted in its present form